# Deep Learning for Magnetic Resonance Fingerprinting

**Solene Girardeau**                                   SOLENE.GIRARDEAU@KCL.AC.UK
**Ilkay Oksuz**                                             ILKAY.OKSUZ@KCL.AC.UK
**Gastao Cruz**                                           GASTAO.CRUZ@KCL.AC.UK
**Claudia Pietro Vasquez**                       CLAUDIA.PIETRO@KCL.AC.UK
**Andrew King**                                        ANDREW.KING@KCL.AC.UK
**James Clough**                                        JAMES.CLOUGH@KCL.AC.UK
*School of Biomedical Engineering & Imaging Sciences, King's College London, UK*

**Editors:** Under Review for MIDL 2019

## Abstract

Standard methods for quantitative MRI are quite time consuming whereas techniques based on deep learning have the potential to be significantly faster while also improving parameter estimation accuracy. The presented models aim to explore the different aspects of MR data, notably the spatial and temporal correlations in and between the signal evolutions. The models developed include purely temporal-focused and spatial-focused models as well as a model trained in both domains. The importance of pre-selecting important features prior to training was also studied and tested.

**Keywords:** Deep Learning, Magnetic Resonance Fingerprinting, Quantitative MRI, Magnetic Resonance Imaging

## 1. Introduction

In recent years a novel approach to quantitative MRI, Magnetic Resonance Fingerprinting (MRF) has been developed. Quantitative MRI aims to measure specific tissue parameters, which can only be seen as contrasts on MRI scans, so as to better characterise biological tissue. In a clinical context, this would allow for a direct comparison between patients and a healthy population, as well as for following the progression of longitudinal diseases (Pierpaoli, 2010). Currently, quantitative MRI is inefficient as it only allows for one parameter to be measured at a time. The sequence must also be carefully constructed, and the signal function must be fitted in each voxel, which means the sequence is much slower than a weighted image. Thus, it is often unsuitable for a clinical environment, requiring very time consuming scans which would be rendered useless by interscan motion. Magnetic Resonance Fingerprinting (MRF) is a recent technique which 'aims at providing simultaneous measurements of multiple parameters such as T1, T2, relative spin density, B0 inhomogeneity (off-resonance frequency), etc., using a single, time-efficient acquisition' (Coppo et al., 2016). MRF works by varying acquisition parameters in a pseudorandom manner so as to get unique, uncorrelated signal evolutions from each of the tissues. These are then compared to a dictionary of simulated signals each with corresponding parameter values which are assigned to the closest matching measured signal evolution. However, this dictionary matching algorithm is quite time inefficient and lacks robustness to noise which has led to

the development of deep learning methods to determine the tissue parameters. Most of these methods take into account either the spatial or the temporal nature of the signal evolution. Studying currently available techniques, we aimed to explore and compare different approaches taking into account both spatial and temporal properties of the MRF signal.

## 2. Methods

All models were trained and tested on a digital brain phantom acquired on a 1.5T Ingenia MR system (Philips, Best, The Netherlands) using a 12-element head coil following the data acquisition developed by Cruz et al (Cruz et al., 2018). The image contrast was simulated at each point in time for a total of 1750 time-points. The code was written using Python 3.6 in Colaboratory, an online Jupyter notebook environment. The models were trained using the deep learning library Keras (Chollet et al., 2015) implemented on the TensorFlow backend (Abadi et al., 2015).

The importance of the temporal property of the data was determined by LSTM-based deep learning architectures inspired by Oksuz et al's work (Oksuz et al., 2018). A basic LSTM model composed only of LSTM layers followed by some fully connected layers was compared to an LSTM-CNN model. The latter was composed of a single LSTM layer followed by 2D convolutional layers. The data for the LSTM-CNN model had to be downsampled (in the time dimension, to 292 time-points) due to memory limitations.

The spatial correlations' importance was studied using CNN-based models. A Dense-CNN model composed of fully connected layers followed by 2D convolutional layers was developed and compared to a CNN model. The latter had the same architecture without the initial fully connected layers. Both models followed a pre-processing of the data which included applying a singular value decomposition (SVD) to extract the most important features. This was applied to each signal in the temporal dimension, originally composed of 1750 time-points, to a number defined by the rank of the SVD. The models were both trained for several SVD ranks, selecting from 1 to 500 important features. The initial Dense-CNN model was then optimised giving the least amount of error for 60 important features (Final Proposed - Table 1).

| Method | Prediction Time (s) | T1 MAE (ms) | T2 MAE (ms) |
|---|---|---|---|
| Dictionary Matching | 96.48 | 266.90±40.42 | 79.72±6.74 |
| Balsiger (Balsiger et al., 2018) | 11.05 | 193.25±34.69 | 53.78±12.71 |
| Cohen (Cohen et al., 2018) | 18.21 | 222.03±12.42 | 72.01±3.10 |
| Hoppe (Hoppe et al., 2017) | **1.51** | 85.79±27.10 | 23.26±10.45 |
| CNN | 2.07 | 81.20±28.13 | 29.96±10.82 |
| LSTM | 59.49 | 101.90±27.98 | 25.89±10.02 |
| Final Proposed | 3.48 | **60.68±22.33** | **16.34±6.09** |

Table 1: Comparison of the T1 and T2 mean absolute error for different methods

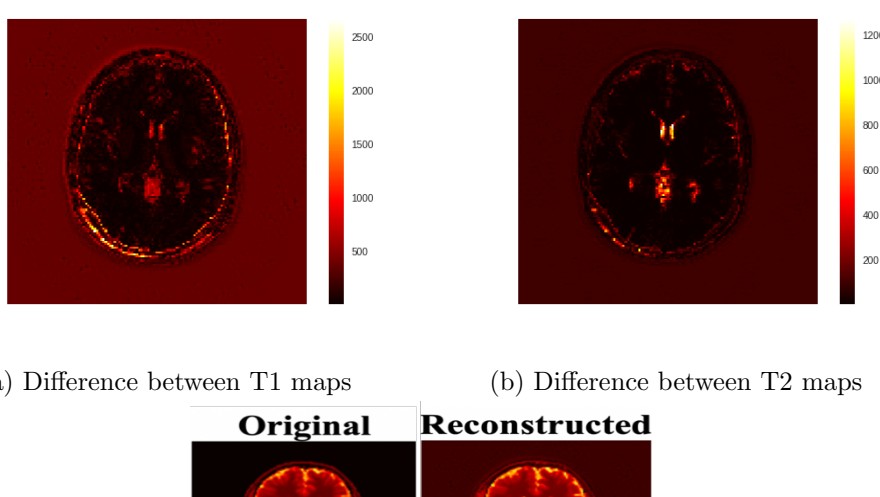

(a) Difference between T1 maps      (b) Difference between T2 maps

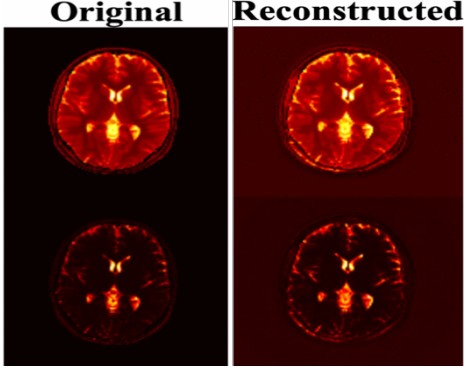

(c) Original and reconstructed T1 (top) and T2 (bottom) maps from the final model predictions

Figure 1: Comparison of T1 and T2 maps

## 3. Conclusions

The LSTM-CNN model achieved errors of 88.84±26.23ms for T1 and 25.62±10.08ms for T2 when the data was downsampled by a factor of 6. The basic LSTM model was found to perform slightly worse than the LSTM-CNN model although no downsampling was required in this case. Prior to optimisation, the Dense-CNN model was found to give the least error after applying an SVD reducing the number of features trained to 60. Errors in this case were found to be 86.96±24.21ms and 26.75±7.99ms for T1 and T2 respectively. The final version of the Dense-CNN model using 60 features was found to be most accurate. Mean absolute errors were 60.68±22.33ms for T1 and 16.34±6.09ms for T2, outperforming other methods. Comparing with the CNN model demonstrated the importance of the initial fully connected layers. These act as pre-feature processors, selecting important features in the data to allow for a more efficient performance of the convolutional layers afterwards.

Future works may include training the LSTM-CNN model with full temporal resolution so as to compare more accurately the performance of the LSTM-CNN model with the other models, and assessing these methods on larger datasets.

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
