# OpenReview forum: "Deep Learning for Magnetic Resonance Fingerprinting"
_MIDL.io/2019/Conference/Abstract — MIDL Abstract 2019_

### Official Review · AnonReviewer2 · 2019-05-02
**Exploration of deep learning models for MRF**

**Rating:** 3
**Confidence:** 2

**Review:**

Pros:
- Nice application of deep learning to an exciting new field, MRF
- Proposed method seems to outperform other works in terms of prediction accuracy and with fast runtime.

Cons:
- The writing is a little hard to understand. It is unclear exactly what is used as input into each of the models, although the reader may be able to infer - it would be good to explicitly state.
- It is unclear how the best SVD rank is chosen - is this based on test error? And what does it mean on p. 3 when the authors say "Prior to optimisation, the Dense-CNN model was found to give the least error after applying an SVD reducing the number of features trained to 60." - the model must have been trained to determine this.
- It would be good if Table 1 included the results for all the proposed architectures in the methods section - i.e. LSTM-CNN is missing.
- Why is data reduction used for CNN models but not LSTMs?

---

### Official Review · AnonReviewer1 · 2019-05-06

**Rating:** 3
**Confidence:** 3

**Review:**

The paper evaluates several deep learning architectures including 3 recently proposed methods and the classical dictionary matching baseline for Magnetic Resonance Fingerprinting. An architecture that outperforms other methods is proposed. The paper is well-written and evaluation is convincing.

---

### Decision · Program_Chairs · 2019-05-06
**Acceptance Decision**

Accept